# Understanding Knowledge Distillation in Post-Training: When It Helps and When It Fails

## Abstract

Large language models (LLMs) achieve strong performance across many tasks, but their high computational cost limits deployment in resource-constrained environments. Knowledge Distillation (KD) offers a practical solution by transferring knowledge from a teacher model of a larger size to a smaller student model. While prior work has mainly examined task-specific or small-scale settings, the post-training stage for building general instruction-following models has received limited attention. In this paper, we conduct a systematic study of KD in post-training using the large-scale Tulu 3 dataset. We find that KD outperforms supervised fine-tuning (SFT) in low-data regimes, but its advantage diminishes as more training data is added. Distilling from a stronger instruction-tuned teacher restores substantial gains even with abundant data, indicating that KD remains effective when the teacher provides knowledge that the student cannot easily acquire from the training data alone. We further study domain-specific, low-resource scenarios and propose a two-stage KD strategy that leverages synthetic teacher-labeled data followed by refinement on human annotations. This method consistently improves student performance, providing practical guidance for building compact models in data-scarce environments.

## 1 Introduction

Large Language Models (LLMs) have brought significant advancements to natural language processing, achieving state-of-the-art performance across a wide range of tasks (OpenAI, 2023; Yang et al., 2025; DeepSeek-AI et al., 2025). However, deploying these models in resource-constrained environments, such as mobile phones and edge devices, remains a considerable challenge due to their high computational and memory demands. To address this issue, model compression techniques, particularly *Knowledge Distillation* (KD), have gained substantial attention as a practical solution for improving efficiency without severely compromising performance.

KD transfers knowledge from a large, over-parameterized *teacher* model to a smaller, more efficient *student* model by encouraging the student to mimic the teacher's output distribution or internal representations (Hinton et al., 2015). This approach allows the student model to achieve competitive performance while significantly reducing resource consumption. Consequently, KD has been widely explored in the context of LLMs, yielding promising results.

Several KD methods have been proposed to enhance its effectiveness for generative language models. SeqKD (Kim & Rush, 2016) encourages the student to imitate the output sequences of the teacher directly. MiniLLM (Gu et al., 2024) replaces the commonly used forward Kullback-Leibler divergence (KLD) with reverse KLD, which is better suited to sequence generation tasks. GKD (Agarwal et al., 2024) introduces a generalized KD framework that supports a range of divergence measures, such as generalized Jensen-Shannon divergence, and reduces train-inference mismatch by incorporating on-policy samples from the student. Most recently, Direct Preference Knowledge Distillation (DPKD) (Li et al., 2024) reformulates KD as a direct preference learning problem, supplementing KL divergence with an implicit reward signal to better align the student with teacher preferences.

While these approaches demonstrate strong performance, they are typically applied in task-specific or small-scale settings. A relatively underexplored but increasingly important scenario is the *post-training* setting, where a student model is trained to acquire general instruction-following capabilities from a teacher model. This setting is particularly relevant for building smaller, more efficient models that can follow human instructions across diverse domains, yet existing work has provided limited insight into the behavior and effectiveness of KD in this context.

In this paper, we conduct a comprehensive study of KD methods applied in the post-training stage of LLM development. We focus on understanding their effectiveness across different training data scales. To this end, we utilize the large-scale instruction-following dataset Tulu 3 (Lambert et al., 2024), which contains 939k high-quality instruction-response pairs. Using this dataset, we train both teacher and student models, and apply KD using subsets of varying sizes.

Our findings reveal that KD provides clear performance benefits over supervised fine-tuning (SFT) in low-data regimes. However, as the size of the training dataset increases, the performance gap between KD and SFT narrows substantially, and KD offers little additional gain. This suggests that KD does not scale effectively to large-data settings, as the student can already recover most of the teacher's knowledge through direct supervision.

To further test this hypothesis, we replace the original teacher model with a stronger, instruction-tuned LLM (e.g., Llama3.3-70B-Instruct) trained on a much larger and more diverse corpus via reinforcement learning from human feedback (RLHF). We find that distillation from this stronger teacher significantly improves the student's performance, even in the large-data setting, highlighting that KD remains effective when the teacher possesses knowledge that the student cannot easily acquire from the data alone.

While our primary focus is on post-training KD for general instruction-following, real-world deployment often involves domain-specific applications, such as translation, summarization, or scientific QA, where high-quality labeled data is scarce. In such cases, models trained with limited supervision are prone to underfitting, and leveraging a stronger teacher becomes particularly valuable. Although KD has been applied in various such contexts, there is a lack of systematic study on its effectiveness across different low-resource domains and the optimal strategies to employ in these data-scarce scenarios.

Motivated by this gap, we further investigate the application of KD in domain adaptation settings. We propose a two-stage training paradigm that first uses teacher-annotated synthetic data to broaden the student's exposure to diverse instruction styles, and then refines the model with KD on the small set of high-quality human annotations. Our experiments show that this strategy consistently improves performance across multiple domain-specific tasks, demonstrating that carefully designed KD pipelines can substantially benefit compact student models in data-scarce environments.

In summary, our contributions are threefold:

- We present a comprehensive study of knowledge distillation in the post-training stage of LLMs, systematically evaluating its effectiveness across different data scales.

- We identify the scaling limitation of KD when the student and teacher are trained on the same dataset, and demonstrate that distilling from a stronger instruction-tuned teacher can still provide substantial benefits.

- We introduce a two-stage KD strategy leveraging synthetic data for domain-specific, low-resource scenarios, which consistently improves student performance and provides practical guidance for real-world deployment.

## 2 RELATED WORK

**Knowledge Distillation and Instruction Tuning.** Knowledge Distillation (KD) transfers knowledge from a large teacher model to a smaller student by matching output distributions or sequences (Hinton et al., 2015). Early methods such as SeqKD (Kim & Rush, 2016) focused on sequence-level imitation, while MiniLLM (Gu et al., 2024) introduced reverse-KL objectives better suited for generation. GKD (Agarwal et al., 2024) addressed train–inference mismatch via on-policy samples, and DPKD (Li et al., 2024) reformulated distillation as preference optimization. Despite their effectiveness, these approaches are typically applied in task-specific or small-scale settings, leaving the post-training stage—critical for building general instruction-following models—underexplored. In

contrast, recent instruction-tuning efforts such as InstructGPT (Ouyang et al., 2022), Alpaca (Taori et al., 2023), OpenAssistant (Köpf et al., 2023), and Tülu 3 (Lambert et al., 2024) highlight the importance of alignment after pretraining, yet rely mostly on supervised fine-tuning or RLHF rather than KD. Our work bridges these directions by systematically studying KD in the post-training stage, evaluating its effectiveness across data scales and highlighting scenarios where stronger teachers remain beneficial.

**Task-Specific KD and Synthetic Data.** Beyond general instruction-following, KD has been widely applied in domain-specific and low-resource scenarios, such as translation, summarization, and QA (Kim & Rush, 2016). More recently, Speculative KD (Xu et al., 2025) improves efficiency and robustness by interleaving student and teacher generation. While these studies confirm the usefulness of KD under limited supervision, they primarily focus on single tasks rather than systematic analysis across domains. Another line of work explores synthetic data as a complementary supervision source: teacher-generated corpora can boost student performance (Shirgaonkar et al., 2024), and even self-training with student generations can yield competitive results (Lewis et al., 2025). However, naive mixing of synthetic and human-annotated data often introduces noise that harms performance. Our work addresses this challenge by proposing a two-stage KD strategy that leverages synthetic data as a warm-up before distillation on gold annotations, demonstrating consistent improvements in domain-specific, low-resource settings.

# 3 POST-TRAINING KNOWLEDGE DISTILLATION FOR LLMS

## 3.1 PROBLEM FORMULATION

We investigate KD in the post-training stage of LLM development, where both the teacher and student are non-intruct tuned language models. The goal is to assess whether KD can effectively transfer general instruction-following capabilities from a teacher to a student when both are fine-tuned on the same dataset.

Formally, let $T$ denote the teacher model and $S$ the smaller student model. Given a dataset $\mathcal{D} = \{(x_i, y_i)\}$ of instruction-response pairs, we compare two training paradigms for the student: (1) supervised fine-tuning (SFT) directly on $\mathcal{D}$, and (2) KD from a teacher $T_s$, which is first trained on $\mathcal{D}$ via SFT. We aim to evaluate whether the student can benefit from distillation beyond what is learned from direct supervision alone, and how this benefit varies with the size of $\mathcal{D}$.

## 3.2 EXPERIMENTAL SETUP

**Dataset and Evaluation Tasks** We conduct experiments using the Tulu 3 dataset (Lambert et al., 2024), which contains 939k high-quality instruction-response pairs. We split this dataset into subsets of varying sizes to evaluate the effectiveness of KD across different data scales. In specific, we create subsets of sizes ranging from 10k samples to the full training set. Note that both the teacher and student models are trained on the same subset.

For evaluation, we use five diverse benchmarks that collectively assess reasoning, scientific knowledge, and instruction-following capabilities: BBH (Srivastava et al., 2023), GPQA (Rein et al., 2023), IFEval (Zhou et al., 2023), InFoBench (Qin et al., 2024), and MMLU-Pro (Wang et al., 2024). These benchmarks cover a wide range of domains, from general reasoning to domain-specific scientific QA and fine-grained instruction following. A detailed description of each benchmark is provided in Appendix A.3.

We report the average accuracy for BBH, GPQA, and MMLU-Pro. For IFEval, we report the prompt-level loose-accuracy, which measures the percentage of prompts for which the model's response satisfies at least one of the constraints specified in the prompt. For InFoBench, we report the Decomposed Requirements Following Ratio (DRFR) (Qin et al., 2024), which measures the percentage of requirements satisfied by the model responses. We use the official evaluation scripts provided by the respective benchmarks to compute these metrics.

**Models** We use the Llama3.1-70B model (Dubey et al., 2024) as our teacher model. To investigate the impact of student model size, we use the Llama3.1-8B, Llama3.2-1B and Llama3.2-3B models as our student models.

We first fine-tune the teacher model on the Tulu 3 dataset using supervised fine-tuning (SFT) to obtain $T_s$. The student models are then trained either via SFT directly on the same dataset, or via knowledge distillation from $T_s$. The training details are listed in Appendix A.2.

### 3.3 KNOWLEDGE DISTILLATION METHOD

According to Ramesh et al. (2025), various knowledge distillation methods do not show significant differences in performance for LLMs. Therefore, we adopt the representative method GKD (Agarwal et al., 2024). GKD is a flexible framework for distilling auto-regressive language models, addressing the train-inference mismatch by incorporating *on-policy* student-generated sequences during training. Unlike standard distillation methods that rely solely on fixed datasets (e.g., ground-truth or teacher-decoded sequences), GKD enables distillation on a mixture of supervised and student-generated data, guided by token-level feedback from the teacher model.

Let $x$ denote an input, $y$ a target sequence, and $\lambda \in [0,1]$ the proportion of on-policy (student-generated) data. The GKD objective is:

$$\mathcal{L}_{\text{GKD}} = (1-\lambda)\,\mathbb{E}_{(x,y)\sim\mathcal{D}}\left[KL(p_T \parallel p_S; y, x)\right] + \lambda\,\mathbb{E}_{x\sim\mathcal{X}}\left[\mathbb{E}_{\hat{y}\sim p_S(\cdot|x)}\left[KL(p_T \parallel p_S; \hat{y}, x)\right]\right], \quad (1)$$

where $KL(p_T \parallel p_S; y, x)$ is the average token-level divergence:

$$KL(p_T \parallel p_S; y, x) = \frac{1}{|y|}\sum_{n=1}^{|y|} KL_{\text{token}}\left(p_T(\cdot \mid y_{<n}, x) \parallel p_S(\cdot \mid y_{<n}, x)\right), \quad (2)$$

and $KL_{\text{token}}$ can be instantiated as forward KL, reverse KL, or generalized Jensen-Shannon divergence.

In particular, the generalized Jensen-Shannon divergence between two distributions $P$ and $Q$ is defined as:

$$\text{JSD}_\beta(P \parallel Q) = \beta \cdot \text{KL}(P \parallel M) + (1-\beta) \cdot \text{KL}(Q \parallel M), \quad \text{where } M = \beta P + (1-\beta)Q. \quad (3)$$

By adjusting the hyperparameter $\beta \in (0,1)$, GKD smoothly interpolates between different divergence behaviors. When $\beta$ approaches 0, $\text{JSD}_\beta$ behaves similarly to forward KL, which is mode-covering: the student must assign probability mass wherever the teacher has support, leading to broader but potentially less precise coverage. Conversely, when $\beta$ approaches 1, $\text{JSD}_\beta$ resembles reverse KL, which is mode-seeking: the student focuses on the teacher's high-probability regions, yielding sharper but less diverse generations. In our experiments, we adopt $\beta = 0.5$ as the default, which balances the two effects and has been shown to perform well in practice (Agarwal et al., 2024).

### 3.4 KD TRAINING PARADIGMS

While the distillation objective specifies *how* knowledge is transferred from the teacher to the student, the initialization of the student model determines *what* prior capabilities it possesses before distillation begins. This initialization choice can substantially influence learning dynamics and final performance, especially under different data regimes. To investigate this factor, we compare two KD training paradigms that differ in whether the student starts from a raw pre-trained checkpoint or from an SFT-adapted model:

- **Base model as student (Base-S)**: The student is initialized with the pre-trained weights of the base model (e.g., Llama3.1-8B) and trained via KD from the teacher model $T_s$.
- **SFTed model as student (SFT-S)**: The student is first fine-tuned on the training dataset via SFT to obtain $S_s$, and then further trained via KD from the teacher model $T_s$.

We evaluate both paradigms on Llama3.1-8B as the student model, varying the training set size from 10k to the full 939k samples. Results in Figure 1 show that the SFT-initialized student consistently outperforms the base-initialized student across most data sizes, though the gap narrows as more data is used. Even with the full dataset, the SFT-initialized student matches or slightly exceeds the base variant, indicating that SFT provides a stronger starting point for KD. This suggests that prior adaptation to instruction-following enables the student to learn more effectively from the teacher.

Based on these results, we adopt the SFT-initialized student as the default configuration for KD in subsequent experiments, as it offers a stronger prior for learning from the teacher.

### 3.5 RESULTS AND ANALYSIS

We report the average performance of student models trained via SFT and KD on five evaluation benchmarks across varying data sizes in Figure 2. In low-data regimes (fewer than 80k samples), KD

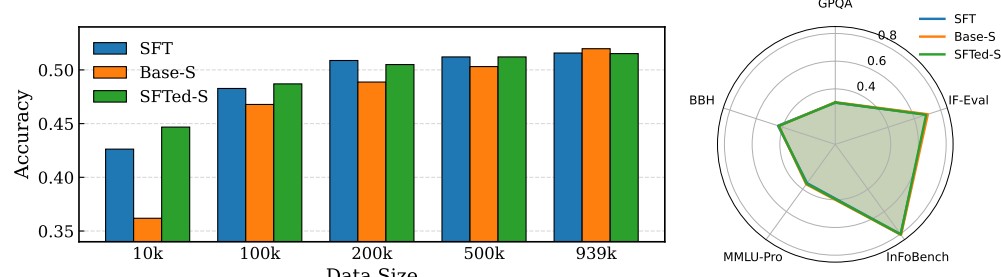

Figure 1: **Left:** Performance of the student model (Llama3.1-8B) trained via SFT and KD under different initialization paradigms across varying training set sizes. **Right:** Task-level performance on the full training set. Across most data scales, the SFT-initialized student outperforms the base-initialized student, with the gap narrowing as data increases. When trained on the full dataset, their performances converge to nearly the same level, indicating that sufficient supervision largely closes the initialization gap. This suggests that prior SFT adaptation offers a stronger starting point for KD.

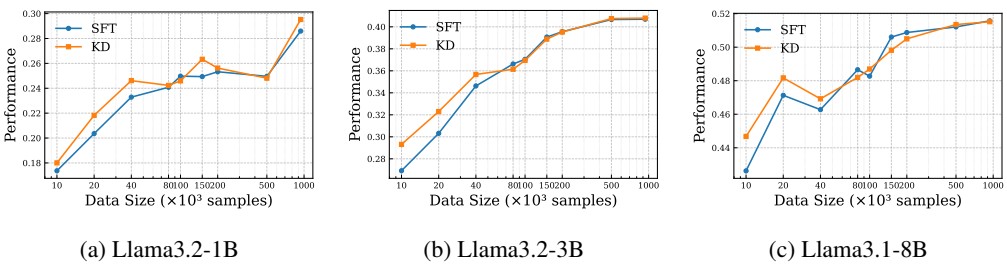

| (a) Llama3.2-1B | (b) Llama3.2-3B | (c) Llama3.1-8B |
|---|---|---|

Figure 2: Performance of three student models (Llama3.2-1B, Llama3.2-3B, Llama3.1-8B) trained with SFT and KD across different training set sizes (logarithmic scale on the x-axis). Results are averaged over five benchmarks: BBH, GPQA, IFEval, InFoBench, and MMLU-Pro. KD provides clear gains in low-data regimes, while the advantage diminishes as more training data is used. Complete numerical results for all models and data sizes are provided in Appendix A.3.

consistently outperforms SFT, with the largest gain of up to 5% absolute observed at 10k samples. This suggests that KD can effectively transfer knowledge from the teacher, enabling more efficient learning when training data is scarce. As the training set grows, the performance gap narrows, and in some cases SFT even surpasses KD (e.g., Llama3.1-8B at 150k samples), indicating that with sufficient data, the student can acquire most of the teacher's knowledge through direct supervision, reducing the benefits of KD.

To examine this further, we replace GKD with SeqKD, a more traditional approach that omits on-policy samples, and train Llama3.1-8B on the full Tulu 3 dataset. For each instance, we sample five outputs from the teacher and train the student to minimize the cross-entropy loss against these sequences. As shown in Table 1, SeqKD achieves similar performance to GKD, confirming that the KD method choice has limited impact when ample training data is available, consistent with prior findings (Ramesh et al., 2025). This supports our hypothesis that, in large-data regimes, little additional information remains to be distilled.

We further hypothesize that this limitation arises because the teacher and student are trained on the same dataset, leaving the student with few opportunities to acquire novel knowledge. In such cases, the supervision provided by KD largely duplicates the ground-truth labels, limiting the marginal utility of distillation. Consistent with this, we observed that when distilling from a same-dataset teacher, the KD loss of the student began at a low value and fluctuated without further reduction, suggesting that the student had little additional signal to absorb. To test this hypothesis, we replace the teacher with a stronger instruction-tuned model, Llama3.3-70B-Instruct (denoted as GKD-IT), which has broader instruction-following capabilities and exposure to more diverse tasks. As shown in Figure 3, GKD-IT substantially outperforms the original GKD teacher, achieving an average performance improvement of around 4% across the five benchmarks. This result supports our intuition:

| Model | BBH | GPQA | IF-Eval | InfoBench | MMLU-pro | Average |
|-------|-----|------|---------|-----------|----------|---------|
| SFT | **43.42** | 30.04 | **69.50** | 80.25 | 34.65 | **51.57** |
| GKD | 42.93 | 30.22 | 68.95 | 80.33 | **35.16** | 51.52 |
| SeqKD | 42.22 | **30.59** | 63.59 | **81.63** | 34.64 | 50.53 |

Table 1: Performance of different methods trained on full Tulu3 training set across five evaluation benchmarks.

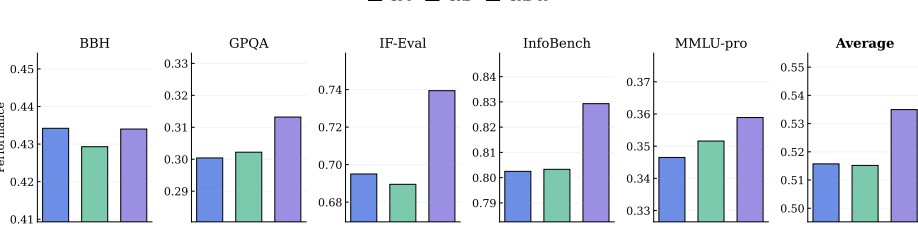

Figure 3: Performance of Llama3.1-8B with GKD and GKD-IT on the full Tulu 3 dataset. Using a stronger instruction-tuned teacher (GKD-IT) yields notable improvements, indicating that distillation from a more capable teacher can still provide substantial benefits.

distillation remains beneficial when the teacher brings in knowledge beyond the training data, enabling the student to learn patterns and reasoning strategies that it would not acquire through direct supervision alone.

**General Takeaway**: Knowledge distillation provides the greatest benefits in low-data regimes, making it particularly valuable for scenarios where only small or domain-specific datasets are available. Moreover, when distilling from a stronger instruction-tuned teacher, substantial gains can still be observed even in large-data settings, indicating that KD remains effective whenever the teacher contributes knowledge beyond the training set.

## 4 TASK-SPECIFIC KNOWLEDGE DISTILLATION

Motivated by the findings in Section 3.5, we examine the use of KD in domain-specific, low-resource settings. Such scenarios are common in real-world applications, where high-quality labeled data for a specialized domain is often scarce. We evaluate the performance of the student model on several domain-specific tasks and compare multiple KD strategies to assess their effectiveness under these constraints.

### 4.1 EXPERIMENTAL SETUP

We use the same student models as in Section 3.5, but now focus on domain-specific tasks. Specifically, we consider the following tasks:

**Low-resource Translation** We adopt the Assamese–English subset of the Flores-200 dataset (Costa-jussà et al., 2022) in the low-resource setting, using the processed data splits provided by (Xu et al., 2025). Specifically, 997 instances from the development set are used for training, while the original test set (1012 instances) is split into 500-instance development and 512-instance test sets. Translation quality is evaluated with the COMET metric (Rei et al., 2022).

**Dialogue Summarization** We use the DialogSum dataset (Chen et al., 2021) following the preprocessed splits in (Xu et al., 2025). The training set contains 1k instances, and evaluation is conducted on the official development set (500 instances) and a 1500-instance test set. Summarization quality is measured using ROUGE-L (Lin, 2004).

**ARC-Challenge** We use the ARC-Challenge dataset (Clark et al., 2018), which consists of multiple-choice grade-school science questions that are deliberately constructed to be difficult for surface-level methods such as retrieval or word co-occurrence. The Challenge Set contains questions that require deeper knowledge and reasoning. Following the processed splits provided by (Ramesh et al., 2025), we use 1.1k instances for training, 500 for development, and 672 for testing. Performance is evaluated using accuracy.

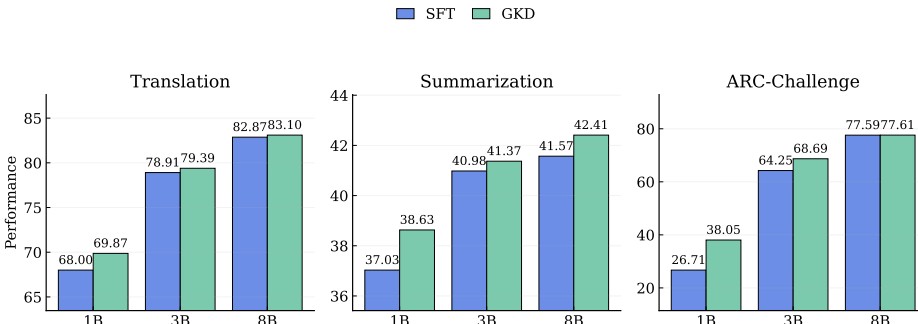

Figure 4: Performance of student models on Translation, Summarization, and ARC-Challenge. GKD improves over SFT across all tasks, but the gains diminish as the student model size increases.

Before doing task-specific training, we first fine-tune the student models on the Tulu 3 dataset via SFT to obtain student models. We then apply knowledge distillation from the teacher model $T_s$ to the student models, using the GKD. The training details are listed in Appendix A.2.

## 4.2 KD RESULTS AND ANALYSIS IN DOMAIN-SPECIFIC TASKS

We report the results of student models trained with SFT and GKD on the three domain-specific tasks in Figure 4. Overall, GKD consistently improves performance compared to SFT, confirming the effectiveness of distillation in these settings. This observation aligns with our earlier findings in Section 3.5, where KD provided the largest benefits in low-data regimes: when task-specific training data is scarce, the student can better leverage the additional knowledge provided by the teacher.

We also observe that the magnitude of improvement varies with student model size. The performance gain from GKD is most pronounced for the 1B student, moderate for the 3B student, and becomes marginal for the 8B student. This trend suggests that smaller models depend more on distillation to acquire knowledge that cannot be fully captured from limited supervision, whereas larger models are capable of learning much of the teacher's knowledge directly from data, leaving less room for additional gains.

In practice, these findings imply that knowledge distillation is particularly valuable for building compact student models that need to operate in domain-specific, data-scarce environments. Such scenarios are common in real-world applications (e.g., specialized translation systems or domain-specific assistants), where the ability to improve small models with limited data is often more crucial than optimizing already strong larger models.

## 4.3 STRENGTHENING KNOWLEDGE DISTILLATION WITH SYNTHETIC DATA

The preceding results suggest that knowledge distillation is most beneficial in low-data regimes, yet its advantages diminish as the amount of available training data increases. One key limitation lies in the heavy reliance on human-annotated data: when the student and teacher are exposed to the same dataset, the student can already recover much of the teacher's knowledge through direct supervision, leaving limited room for further gains. In practice, labeled data in specialized domains is scarce, and scaling up high-quality annotation is often impractical. To address this challenge, we propose to augment KD with synthetic data.

**Synthetic Data Generation.** Concretely, we employ the Llama3.1-70B model fine-tuned on Tulu 3 as the generator $T_{\text{gen}}$. Given a small set of demonstrations $\{(x_j, y_j)\}_{j=1}^k$ sampled from the training data $\mathcal{D}$ and an instruction prompt $I$[1], we construct the in-context prompt

$$\texttt{Prompt} = I \oplus \big[\texttt{In:}\quad x_1\ \texttt{Out:}\quad y_1\ \ldots\ \texttt{In:}\quad x_k\ \texttt{Out:}\quad y_k\ \texttt{In:}\big], \quad (4)$$

where $\oplus$ denotes concatenation. Conditioned on this prompt, the generator samples an unlabeled input

$$x^s \sim T_{\text{gen}}(\cdot \mid \texttt{Prompt}). \quad (5)$$

---

[1]Instruction for unlabeled data generation is provided in Appendix A.4

| Model | Translation | | | | Summarization | | | | ARC-Challenge | | | |
|---|---|---|---|---|---|---|---|---|---|---|---|---|
| | SFT | GKD | Mix | 2-Stage | SFT | GKD | Mix | 2-Stage | SFT | GKD | Mix | 2-Stage |
| Teacher (70B SFT) | 86.31 | – | – | – | 44.99 | – | – | – | 91.38 | – | – | – |
| 1B | 68.00 | 69.87 | 78.13 | **78.65** | 37.03 | 38.63 | 40.48 | **42.84** | 26.71 | 38.05 | 70.22 | **70.31** |
| 3B | 78.91 | 79.39 | 82.41 | **82.67** | 40.98 | 41.37 | 42.17 | **44.02** | 64.25 | 68.69 | 80.22 | **80.46** |
| 8B | 82.87 | 83.10 | 83.15 | **83.63** | 41.57 | 42.41 | 43.40 | **44.29** | 77.99 | 77.05 | 85.35 | **85.67** |

Table 2: Performance of different models on Translation, Summarization, and ARC-Challenge. We compare SFT, KD, Mix (GKD with synthetic data mixing), and 2-Stage (two-stage GKD). Best results for each model are highlighted in **bold**.

Repeating this process yields a synthetic unlabeled set $\mathcal{X}^s = \{x_1^s, \ldots, x_N^s\}$. These inputs are subsequently annotated by the teacher model $T$, which is the Llama3.1-70B fine-tuned on the corresponding training subset $\mathcal{D}$, to obtain pseudo-labeled pairs

$$\mathcal{D}^s = \{(x_i^s, y_i^s)\}_{i=1}^N, \quad y_i^s \sim T(\cdot \mid x_i^s). \tag{6}$$

**Synthetic Data Integration.** A straightforward approach is to simply combine the synthetic dataset $\mathcal{D}^s$ with the original human-annotated dataset $\mathcal{D}$ and train the student under the KD objective:

$$\mathcal{L}_{\mathrm{mix}} = \mathbb{E}_{(x,y)\sim\mathcal{D}\cup\mathcal{D}^s} \left[ \mathrm{KL}(p_T(\cdot \mid x) \parallel p_S(\cdot \mid x)) \right]. \tag{7}$$

As we will show in Section 4.4, this simple mixing strategy indeed provides improvements over SFT, indicating that synthetic data can be beneficial despite its noise However, its effect remains limited, as the noisy synthetic samples $\mathcal{D}^s$ dilute the high-quality supervision from $\mathcal{D}$, preventing the student from fully exploiting the additional data. This motivates the development of a more structured integration scheme.

To this end, we adopt a two-stage training paradigm. In the first stage, the student $S$ is exposed only to the synthetic dataset, performing an initial training step:

$$S^{(0)} = \mathrm{Train}(S; \mathcal{D}^s), \tag{8}$$

where the objective is the standard cross-entropy loss with pseudo-labels $y_i^s$. Although $\mathcal{D}^s$ is noisy, this stage helps the student adapt to a broader distribution of instruction formats and response structures, providing a better starting point.

In the second stage, we initialize the student with $S^{(0)}$ and then optimize the GKD objective on the human-annotated dataset $\mathcal{D}$ (see Sec. 3.3 for details):

$$S^\star = \arg\min_S \mathcal{L}_{\mathrm{GKD}}(S; T, \mathcal{D}) \quad \text{with initialization } S \leftarrow S^{(0)}. \tag{9}$$

This staged design leverages synthetic data as a "warm-up" that broadens the student's exposure, while preserving the high-quality guidance of the teacher on $\mathcal{D}$ in the final distillation step. Empirically, we find that this method significantly improves performance compared to both direct mixing and vanilla KD.

### 4.4 RESULTS OF SYNTHETIC DATA AUGMENTATION

We empirically set the number of synthetic samples $N$ for each task to 100k. Besides SFT and GKD, we report the results of the following two strategies:

- **Mixing**: Directly combining the synthetic dataset $\mathcal{D}^s$ with the original dataset $\mathcal{D}$ and training the student via GKD on the mixed dataset.

- **Two-stage**: The proposed two-stage training paradigm, where the student is first trained on the synthetic dataset $\mathcal{D}^s$, and then fine-tuned via GKD on the original dataset $\mathcal{D}$.

We report the results in Table 2, from which we draw the following findings:

**Mixing synthetic data yields clear improvements.** Across all tasks and model sizes, the mixing strategy consistently outperforms SFT, showing that synthetic data can effectively enhance KD despite its noisiness. For example, the 3B model on *Translation* improves from 78.91 (SFT) to 82.41 with mixing, and the 8B model on *Summarization* improves from 41.57 (SFT) to 43.40 with mixing. These results indicate that even a direct combination of human and synthetic data provides noticeable gains.

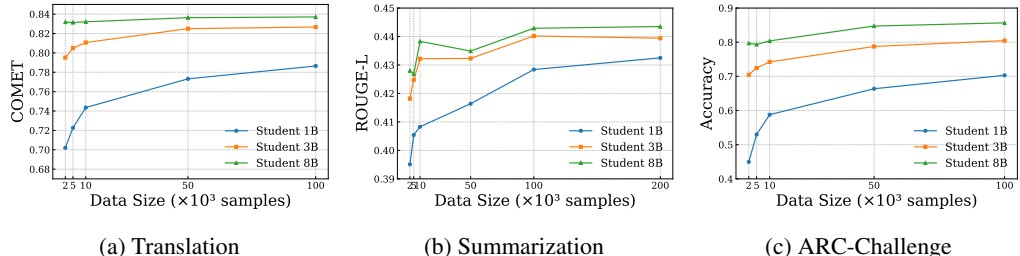

(a) Translation         (b) Summarization        (c) ARC-Challenge

Figure 5: Impact of synthetic data size on knowledge distillation. Performance improves as more synthetic data is added, with smaller models benefiting more, but the gains quickly saturate as data size grows.

**Two-stage training maximizes the benefit of synthetic data.** The two-stage paradigm consistently achieves the best results across all settings. For example, the 1B model on *ARC-Challenge* improves to **44.97**, compared to 26.71 with SFT and 28.66 with mixing. This demonstrates that, while synthetic data is inherently noisy, it becomes substantially more beneficial when used as a warm-up stage before distillation on human-annotated data.

**General takeaway.** Overall, our results show that synthetic data is indeed useful for improving KD in low-resource, domain-specific tasks. Even the simple mixing strategy brings consistent gains over SFT. At the same time, the two-stage paradigm further amplifies these benefits by structuring how synthetic data is leveraged. This demonstrates that synthetic data, whether used directly or in a staged manner, can substantially enhance student models in low-resource settings, with two-stage training providing the most effective integration.

### 4.5 IMPACT OF SYNTHETIC DATA SIZE

To better understand the role of synthetic data, we investigate how the size of the synthetic dataset influences distillation performance. Specifically, we vary the number of synthetic samples $N$ from 5k to 100k for all 1B, 3B, and 8B student models, and report the results in Figure 5. We find that adding synthetic data consistently improves performance across different tasks, with the 1B student benefiting the most. However, the improvement is most significant when moving from very limited to moderate amounts of synthetic data, after which the marginal gains diminish. This indicates that while synthetic data is an effective complement to KD in low-resource settings, its utility does not scale linearly with quantity. Instead, the main advantage comes from a relatively small but diverse synthetic set that broadens the student's exposure beyond what human annotations alone can provide.

## 5 CONCLUSION

In this work, we conducted a systematic study of KD in the post-training stage of LLMs. Through extensive experiments on the large-scale Tulu 3 dataset, we found that KD consistently outperforms SFT in low-data regimes, but its benefits diminish as training data grows. Nevertheless, distilling from a stronger instruction-tuned teacher restores substantial gains even in high-data settings, highlighting that KD remains effective when the teacher possesses knowledge beyond the training set.

We further explored domain-specific, low-resource scenarios and demonstrated that KD is particularly valuable for smaller student models. To address the limitations of scarce human annotations, we introduced a two-stage KD paradigm that first leverages synthetic teacher-labeled data before refining on human annotations. This method consistently improved student performance, surpassing both direct mixing and standard KD, thereby offering a practical recipe for building compact and capable models under resource constraints.

Overall, our findings provide a clearer understanding of when and why KD is most effective, establish its limitations when teacher and student share the same supervision, and present a simple yet effective two-stage framework for integrating synthetic data. We hope these insights will guide future research on efficient LLM post-training and inform the development of deployable models in real-world low-resource applications.

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

# A   APPENDIX

## A.1   DETAILS OF EVALUATION TASKS

- **BIG-bench(BBH)** (Srivastava et al., 2023): A large-scale benchmark comprising 204 diverse tasks spanning linguistics, reasoning, math, science, social bias, and more. Designed to probe capabilities believed to be beyond current language models, it evaluates both quantitative performance and qualitative behaviors across a wide range of domains.

- **GPQA** (Rein et al., 2023): A challenging multiple-choice benchmark of 448 questions in biology, physics, and chemistry, authored by domain experts. The questions are designed to be "Google-proof" and extremely difficult, with expert-level accuracy around 65% and GPT-4 baselines achieving only 39%, making it suitable for evaluating advanced reasoning in specialized scientific domains.

- **IFEval** (Zhou et al., 2023): An instruction-following benchmark of around 500 prompts containing verifiable constraints, such as word-count limits or required keywords. It provides a reproducible and objective way to assess LLMs' ability to follow natural language instructions without relying on costly human evaluation.

- **InFoBench** (Qin et al., 2024): A benchmark of 500 diverse instructions decomposed into 2,250 fine-grained requirements, designed to evaluate LLMs' instruction-following ability under the Decomposed Requirements Following Ratio (DRFR) metric. It enables detailed assessment of compliance with multiple constraint categories and supports evaluation using human or LLM-based annotators.

- **MMLU-Pro** (Wang et al., 2024): A benchmark built upon the Massive Multitask Language Understanding (MMLU) benchmark, which tests language understanding and reasoning across a wide range of subjects. While MMLU mainly contains knowledge-driven multiple-choice questions, MMLU-Pro increases difficulty by replacing trivial or noisy items with reasoning-focused questions and expanding the choice set from four to ten options. This design better discriminates between advanced LLMs and reduces score sensitivity to prompt variations.

## A.2 IMPLEMENTATION DETAILS

We adopt the GKD trainer from the TRL library for all KD experiments. We use the AdamW optimizer with a learning rate of $5 \times 10^{-6}$ and a batch size of 128. The models are trained for 2 epochs. We use a linear learning rate scheduler with a warm-up phase of 3% of the total training steps. The maximum sequence length is set to 4096 tokens. All experiments are conducted on NVIDIA H100 GPUs. To aviod the instability of gradient accumulation [2], we use sum instead of average to compute the loss over multiple batches.

We set the hyperparameter $\lambda$ in Eqn. 2 to 0.5, balancing the contributions of the training data and the on-policy samples. The hyperparameter $\beta$ in the generalized Jensen-Shannon divergence is set to 0.5, which balances the mode-covering and mode-seeking behaviors.

For SFT training, we use the same hyperparameters as in KD training.

For generating synthetic data, we randomly sample 10 examples from the training set as in-context demonstrations. We use nucleus sampling with $p = 1.0$ and a temperature of 0.8 to generate synthetic inputs. The maximum generation length is set to 4096 tokens. We set the temperature to 0.6 when the teacher annotates the synthetic inputs.

## A.3 DETAILED EXPERIMENTAL RESULTS

Table 3 reports the detailed performance of all student models across different training set sizes, complementing the averaged trends shown in Figure 2. Each entry corresponds to accuracy (or task-specific metric) on the five evaluation benchmarks: BBH, GPQA, IFEval, InFoBench, and MMLU-Pro.

The results confirm that knowledge distillation (GKD) yields clear gains over supervised fine-tuning (SFT) in low-data regimes, particularly for smaller students such as Llama3.2-1B and Llama3.2-3B. However, as the training set size increases, the advantage of GKD diminishes, and in some cases SFT matches or slightly surpasses GKD (e.g., Llama3.1-8B at 939k samples). These detailed results provide quantitative evidence for the scaling limitations of KD and support our conclusion that its primary benefits lie in low-resource settings.

## A.4 SYNTHETIC DATA GENERATION PROMPT

The instruction prompt used for generating synthetic data is as follows:

```
You are a data generation assistant.  You will be
given 10 demonstrations
Task:  Based on these examples, produce exactly
one new input message that matches the same task,
language, and style.
Strict requirements:  Output only the message content
itself.  Do NOT include any explanations, quotes,
labels, or the answer.

Here are 10 demonstrations:
<10 examples from the training set>
Now, generate exactly one brand-new input message that
follows the same task and formatting.
```

---

[2] https://unsloth.ai/blog/gradient

```
Output only the input message content itself.  Do not
output any answer or extra text.
```

## B   LLM USAGE

In preparing this paper, we used GPT5 and Gemini solely as a writing assistant to polish the writing. Specifically, LLMs were employed to improve the fluency, clarity, and grammar of the text, while the core research ideas, experimental design, implementation, analysis, and conclusions were entirely developed by the authors.

| Model | Data Size | Method | BBH | GPQA | IF-Eval | InfoBench | MMLU-Pro | Avg. |
|---|---|---|---|---|---|---|---|---|
| llama3.2-1b | 10k | SFT | 2.95 | 25.09 | 12.38 | 34.25 | 12.17 | 17.368 |
| | | GKD | 3.84 | 24.18 | 15.71 | 34.15 | 12.13 | 18.002 |
| | 20k | SFT | 11.06 | 24.73 | 15.90 | 37.84 | 12.25 | 20.356 |
| | | GKD | 14.88 | 25.27 | 17.93 | 38.68 | 12.37 | 21.826 |
| | 40k | SFT | 14.59 | 23.81 | 21.44 | 44.06 | 12.53 | 23.286 |
| | | GKD | 15.02 | 25.09 | 24.77 | 45.70 | 12.48 | 24.612 |
| | 80k | SFT | 13.64 | 25.46 | 23.84 | 45.10 | 12.36 | 24.080 |
| | | GKD | 10.81 | 25.09 | 24.58 | 48.00 | 12.67 | 24.230 |
| | 100k | SFT | 13.75 | 26.19 | 25.69 | 46.65 | 12.58 | 24.972 |
| | | GKD | 11.96 | 23.44 | 27.54 | 47.41 | 12.61 | 24.592 |
| | 150k | SFT | 13.79 | 26.01 | 25.69 | 46.83 | 12.36 | 24.936 |
| | | GKD | 13.84 | 26.92 | 27.73 | 50.64 | 12.52 | 26.330 |
| | 200k | SFT | 13.21 | 26.01 | 26.99 | 48.09 | 12.33 | 25.326 |
| | | GKD | 11.37 | 26.37 | 27.36 | 50.40 | 12.61 | 25.622 |
| | 500k | SFT | 9.32 | 23.81 | 30.13 | 49.05 | 12.48 | 24.958 |
| | | GKD | 4.72 | 24.54 | 30.31 | 51.39 | 13.10 | 24.812 |
| | 939k | SFT | 13.85 | 27.84 | 36.60 | 52.03 | 12.66 | 28.596 |
| | | GKD | 14.02 | 25.64 | 39.93 | 55.11 | 12.92 | 29.524 |
| llama3.2-3b | 10k | SFT | 0.35 | 29.30 | 27.73 | 51.73 | 25.48 | 26.918 |
| | | GKD | 3.96 | 28.39 | 29.21 | 59.05 | 25.96 | 29.314 |
| | 20k | SFT | 12.70 | 26.74 | 33.09 | 53.44 | 25.61 | 30.316 |
| | | GKD | 16.43 | 27.11 | 35.12 | 56.55 | 26.28 | 32.298 |
| | 40k | SFT | 23.22 | 26.92 | 37.71 | 60.36 | 24.92 | 34.626 |
| | | GKD | 22.68 | 27.84 | 39.37 | 62.52 | 25.94 | 35.670 |
| | 80k | SFT | 24.57 | 28.02 | 39.74 | 65.36 | 25.41 | 36.620 |
| | | GKD | 20.93 | 26.92 | 41.04 | 65.70 | 26.13 | 36.144 |
| | 100k | SFT | 23.67 | 26.74 | 43.44 | 66.92 | 24.46 | 37.046 |
| | | GKD | 21.01 | 26.56 | 44.55 | 67.93 | 24.68 | 36.946 |
| | 150k | SFT | 25.03 | 28.39 | 47.69 | 69.17 | 25.07 | 39.070 |
| | | GKD | 21.76 | 29.30 | 48.61 | 69.39 | 25.36 | 38.884 |
| | 200k | SFT | 27.12 | 28.21 | 48.98 | 68.50 | 24.97 | 39.556 |
| | | GKD | 21.55 | 29.30 | 51.76 | 69.23 | 25.79 | 39.526 |
| | 500k | SFT | 25.93 | 25.46 | 55.08 | 73.27 | 23.58 | 40.664 |
| | | GKD | 21.07 | 25.64 | 57.30 | 75.24 | 24.51 | 40.752 |
| | 939k | SFT | 21.30 | 27.11 | 57.30 | 73.79 | 23.96 | 40.692 |
| | | GKD | 18.14 | 27.66 | 58.60 | 74.86 | 24.71 | 40.794 |
| llama3.1-8b | 10k | SFT | 21.98 | 29.85 | 48.98 | 73.90 | 38.41 | 42.624 |
| | | GKD | 25.50 | 31.14 | 51.76 | 76.27 | 38.71 | 44.676 |
| | 20k | SFT | 38.00 | 31.32 | 53.05 | 75.76 | 37.52 | 47.130 |
| | | GKD | 34.74 | 32.05 | 57.49 | 77.47 | 39.09 | 48.168 |
| | 40k | SFT | 31.72 | 30.95 | 55.08 | 76.01 | 37.61 | 46.274 |
| | | GKD | 28.23 | 32.05 | 58.96 | 77.13 | 38.23 | 46.920 |
| | 80k | SFT | 33.94 | 33.15 | 59.52 | 79.29 | 37.38 | 48.656 |
| | | GKD | 28.29 | 31.68 | 63.03 | 79.56 | 38.42 | 48.196 |
| | 100k | SFT | 30.56 | 31.68 | 64.51 | 78.47 | 36.14 | 48.272 |
| | | GKD | 28.54 | 30.77 | 68.76 | 78.86 | 36.56 | 48.698 |
| | 150k | SFT | 40.15 | 31.50 | 65.62 | 79.10 | 36.65 | 50.604 |
| | | GKD | 34.43 | 29.12 | 68.39 | 79.45 | 37.70 | 49.818 |
| | 200k | SFT | 41.47 | 29.85 | 68.58 | 78.58 | 35.90 | 50.876 |
| | | GKD | 39.32 | 29.12 | 68.02 | 78.90 | 37.13 | 50.498 |
| | 500k | SFT | 40.98 | 30.95 | 69.13 | 79.24 | 35.74 | 51.208 |
| | | GKD | 38.77 | 30.04 | 69.50 | 82.64 | 35.80 | 51.350 |
| | 939k | SFT | 43.42 | 30.04 | 69.50 | 80.25 | 34.65 | 51.572 |
| | | GKD | 42.93 | 30.22 | 68.95 | 80.33 | 35.16 | 51.518 |

Table 3: Detailed performance of student models (Llama3.2-1B, Llama3.2-3B, and Llama3.1-8B) trained with SFT and GKD across different training set sizes. Results are reported on five benchmarks (BBH, GPQA, IFEval, InFoBench, and MMLU-Pro).