# OpenReview forum: "Understanding Knowledge Distillation in Post-Training: When It Helps and When It Fails"
_ICLR.cc/2026/Conference — ICLR 2026 Conference Withdrawn Submission_

### Official Review · Reviewer_jcUp · 2025-10-27

**Soundness:** 2
**Presentation:** 2
**Contribution:** 1
**Rating:** 2
**Confidence:** 4

**Summary:**

This paper investigates the effectiveness of knowledge distillation in the post-training phase of LLMs, specifically for building general instruction-following capabilities. The authors conduct a systematic study comparing KD against standard SFT across various training data scales using the Tulu 3 dataset. Their primary findings are threefold: 1) KD significantly outperforms SFT in low-data regimes, but this advantage diminishes as the amount of training data increases; 2) the performance gains from KD can be restored even with abundant data if a much stronger, OOD teacher model is used, suggesting KD is most effective when the teacher possesses knowledge unavailable in the student's training set.

**Strengths:**

1. The research question—"When does KD help and when does it fail?"—is fundamental for practitioners aiming to build smaller, efficient models. The authors design clear, well-controlled experiments that directly test their hypotheses, such as varying the data size to show the scaling limits of KD (Fig. 2) and then introducing a stronger teacher to demonstrate a solution (Fig. 3).

2. The evaluation employs multiple student model sizes (1B, 3B, 8B) and a diverse suite of modern benchmarks (BBH, GPQA, IFEval, etc.), which adds significant credibility to the findings.

**Weaknesses:**

1. The paper's methodological novelty is limited. The core techniques employed—GKD for distillation and in-context learning for synthetic data generation—are existing methods. The paper's contribution is in the analysis and application of these methods, not in advancing the methods themselves. A more critical weakness is the somewhat superficial treatment of different KD methods. The authors conclude from one experiment on a large dataset (Tab. 1) that the choice of KD method has limited impact, but they fail to investigate if this holds true in the low-data regime, which is precisely where they claim KD is most valuable.

2. The concept of a "stronger teacher" could be more precisely defined; the key factor appears to be access to knowledge external to the student's training data, rather than just model size or strength in the abstract. Currently, only the 70B version of the LLaMA family is taken as the teacher model. I do not think this is enough for the statement of "Distilling from a stronger instruction-tuned teacher restores substantial gains even with abundant data, indicating that KD remains effective when the teacher provides knowledge that the student cannot easily acquire from the training data alone.".

3. The proposed two-stage synthetic data strategy, while effective, is a relatively straightforward application of data augmentation; a deeper analysis comparing it to other integration schemes or exploring the quality-quantity trade-off of synthetic data more rigorously would have strengthened this contribution. At present, all experiments in the paper are compared with different stages within their own process, and are not compared with various cutting-edge knowledge distillation solutions.

**Questions:**

I would recommend that the authors conduct additional experiments on more model families (even across families), different teacher models, and cutting-edge knowledge distillation schemes to demonstrate the effectiveness of the proposed method. I would also recommend demonstrating the effectiveness of knowledge distillation in mathematics (e.g., relatively simple gsm8k and MATH) and code (e.g., relatively simple MBPP).

---

### Official Review · Reviewer_ZRn3 · 2025-10-31

**Soundness:** 2
**Presentation:** 2
**Contribution:** 2
**Rating:** 4
**Confidence:** 3

**Summary:**

This paper finds that Knowledge Distillation (KD) for instruction-tuning LLMs is most effective in low-data regimes. This advantage over standard fine-tuning (SFT) disappears as data increases, unless a stronger teacher model (with knowledge beyond the training data) is used. For low-resource domains, a two-stage KD strategy (synthetic data followed by human-annotated data) is highly effective.

**Strengths:**

1. The paper addresses a gap in previous research (which focused on small-scale or task-specific settings) by conducting a systematic study of knowledge distillation in the post-training stage using the large-scale Tulu 3 dataset.
2. It provides a clear, actionable solution for a common problem by proposing a novel two-stage KD strategy (using synthetic data followed by human annotations) that "consistently improves student performance" in data-scarce, domain-specific scenarios.

**Weaknesses:**

1. The finding that "distilling from a stronger instruction-tuned teacher can still provide substantial benefits" is not surprising. Prior work has already established that a stronger teacher model generally leads to better distillation performance.
2. The experimental setup using Llama3 1B and 3B models is potentially flawed. These models are reportedly already distilled, and the teacher model is also from the Llama3 family. The study should have used non-distilled base models to avoid these confounding variables.
3. The method of using in-context learning to generate synthetic data is not novel; this approach was previously established by methods such as SelfInstruct[1].


[1] Self-instruct: Aligning language models with self-generated instructions.

**Questions:**

N/A

---

### Official Review · Reviewer_KW2q · 2025-11-03

**Soundness:** 1
**Presentation:** 2
**Contribution:** 1
**Rating:** 2
**Confidence:** 4

**Summary:**

This paper investigates the effectiveness of knowledge distillation (KD) during the instruction-tuning phase of LLMs. The authors re-confirm the established finding that KD outperforms supervised fine-tuning (SFT) in low-data regimes. Conversely, they find this advantage disappears when training on a large-scale dataset where the teacher and student are trained on the same data, an unsurprising result if teacher and student lack of a knowledge gap. The paper shows this gain can be recovered by using a stronger, externally-trained teacher, though it fails to analyze this effect in depth. Finally, it proposes a simple two-stage data augmentation strategy for low-resource domain adaptation, which involves training on synthetic teacher-labeled data before distilling on the target human-annotated data using on-policy.

**Strengths:**

1. Studying data scaling of KD approaches is an important research problem.

**Weaknesses:**

This paper suffers from several major weaknesses that question its contribution and novelty:

If I understand it correctly, the paper's finding that KD fails in the high-data regime  is unsurprising. If the student can achieve performance similar to the teacher via SFT on the same large dataset, it implies there is no significant knowledge gap left to distill. The experiment simply confirms that KD is ineffective when there is nothing left to transfer, which is not a strong finding. The right setup should be using LLama3.3-70B as a teacher and perform the experiment.

The observation that KD outperforms SFT in low-data scenarios is an established finding in the knowledge distillation literature (e.g., in works like SKD). This paper's confirmation of this dynamic does not constitute a new contribution.

Limited Analysis of KD Methods: The paper's conclusions about KD are based almost entirely on a single, on-policy method (GKD). This is a very narrow scope, and it is unclear if these findings would hold for other prominent KD techniques, such as supervised KD and SKD.

Insufficient Analysis of "Stronger Teacher" Gains: The paper's most interesting finding—that a stronger teacher restores KD's benefits—is left critically under-analyzed. The hypothesis that the teacher provides "knowledge that the student cannot easily acquire from the data alone" is speculative. The paper provides no quantitative or qualitative analysis to show what this extra knowledge is (e.g., improved reasoning, better instruction following, style adherence) or how it is being transferred.

Two-Stage Method as Simple Data Augmentation: The proposed two-stage "improvement" appears to be a straightforward data augmentation strategy rather than a novel KD paradigm. The intuition of familiarizing a student with teacher-generated samples before on-policy KD is similar to prior work (SKD, https://arxiv.org/abs/2410.11325 and DistiLLM https://arxiv.org/abs/2402.03898). The paper fails to provide a necessary comparison of this technique to existing methods that also leverage synthetic data and staged training for bridging the teacher-student gap. The paper does have study on performance regarding to synthetic data size. However, paper should go deeper about what aspects help here: 1) diversity of the prompt? for example.

**Questions:**

1. What is teacher's performance in Sec 3.5 analysis

---

### Official Review · Reviewer_JQ7A · 2025-11-06

**Soundness:** 2
**Presentation:** 3
**Contribution:** 2
**Rating:** 4
**Confidence:** 2

**Summary:**

The paper presented a comprehensive empirical study of knowledge distillation in the post-training stage of LLMs. The authors conducted experiments on the Tulu 3 dataset and found that KD consistently outperforms SFT in low-data regimes. Moreover, in specific scenarios where high-quality datasets are scarce, the experiments demonstrate that KD provides a clear advantage for smaller student models. Finally, to address the shortage of human-labeled data, the authors propose a two-stage KD paradigm that first pre-trains the student using synthetic teacher-labeled data, followed by refinement with human-annotated data.

**Strengths:**

1. The authors conducted extensive and thorough experiments to investigate the behavior of knowledge distillation in the post-training stage of LLMs, examining the effects of teacher model strength, student training paradigm (SFT vs. KD), dataset size, and the use of synthetic data. The experiments are comprehensive and rigorously designed.

2. The paper is well-structured and logically progressive. The authors present their analyses and reflections on the experimental phenomena with clarity and sound reasoning.

**Weaknesses:**

1. The paper primarily provides empirical conclusions based on observed experimental phenomena, but lacks deeper theoretical analysis and discussion to support these findings.

2. Although the authors propose a two-stage distillation method, it appears to have limited novelty — essentially pretraining on synthetic data followed by distillation on high-quality human-labeled data. This approach seems tailored for scenarios with limited human annotations, but the paper would benefit from a more detailed discussion on the applicable data scale for this method. In addition, it would be valuable to analyze the optimal ratio between human-labeled and synthetic data that leads to practical improvements in distillation. The authors could also elaborate on whether they considered the diversity or quality metrics of the synthetic data during generation. Furthermore, including an ablation study to evaluate the contribution of each stage would strengthen the experimental section. It would also be interesting to explore whether reversing the order—training first on human data and then on synthetic data—would affect the results.

3. The comparison methods in the experimental section are relatively limited. It would enhance the paper’s completeness to include comparisons with more recent or diverse distillation approaches.

**Questions:**

1. Could the authors provide more discussion on the dataset scale at which the proposed two-stage distillation method is most effective?

2. What is the appropriate ratio between human-labeled data and synthetic data that yields practical improvements in distillation performance?

3. During synthetic data generation, did the authors consider data diversity or use any specific metrics to evaluate the quality of the generated samples?

4. Could the authors include an ablation study to assess the actual contribution of each stage in the distillation process? Additionally, would reversing the order—training first on human data and then on synthetic data—affect the results?

5. Can the authors compare their method with a broader range of existing distillation approaches to strengthen the experimental validation?

---

### Note · Authors · 2025-11-21

I have read and agree with the venue's withdrawal policy on behalf of myself and my co-authors.